# Association between Daily Activities and Behavioral and Psychological Symptoms of Dementia in Community-Dwelling Older Adults with Memory Complaints by Their Families

**DOI:** 10.3390/ijerph17186831

**Published:** 2020-09-18

**Authors:** Yuriko Ikeda, Gwanghee Han, Michio Maruta, Maki Hotta, Eri Ueno, Takayuki Tabira

**Affiliations:** 1Graduate School of Health Science, Kagoshima University, Kagoshima 890-8544, Japan; 2Department of Neuropsychiatry, Kumamoto University Hospital, Kumamoto 860-8556, Japan; hans11057@gmail.com; 3Doctoral Program of Clinical Neuropsychiatry, Graduate School of Health Science, Kagoshima University, Kagoshima 890-8544, Japan; m.maru0111@gmail.com; 4Department of Rehabilitation, Medical Corporation, Sanshukai, Okatsu Hospital, Kagoshima 890-0067, Japan; 5Department of Behavioral Neurology and Neuropsychiatry, Osaka University United Graduate School of Child Development, Osaka 565-0871, Japan; komaki@psy.med.osaka-u.ac.jp; 6Department of Rehabilitation, Medical Corporation, Nissyoukai, Minamikagoshimasakura Hospital, Kagoshima 890-0069, Japan; eueno@nissyoukai.or.jp

**Keywords:** activities of daily living, behavioral and psychological symptoms of dementia, community-dwelling older adults, caregiver, memory complaints

## Abstract

It is important and useful to consider information provided by family members about individuals with memory complaints’ instrumental activities of daily living (IADL). The purpose of this study was to clarify the characteristics and relevance of individuals with memory complaints’ IADL and behavioral and psychological symptoms of dementia (BPSD) assessed from the perspective of the family members using the Process Analysis of Daily Activity for Dementia and short version Dementia Behavior Disturbance scale. A self-administered questionnaire was sent to 2000 randomly selected members of Consumer’s Co-operative Kagoshima, and 621 responded. Of the returned responses, there were 159 participants who answered about individuals with memory complaints. The stepwise multiple regression analysis was used to examine the association between IADL and BPSD. The result showed that many IADL of the individuals with memory complaints were associated with BPSD of apathy, nocturnal wakefulness, and unwarranted accusations, adjusted for age, gender, and the observation list for early signs of dementia. In addition, each IADL was associated with BPSD of apathy, nocturnal wakefulness, and dresses inappropriately. Modifying lifestyle early on when families recognize these changes may help maintain and improve the long-term quality of life of the individuals with memory complaints and their family.

## 1. Introduction

Individuals suspected to have dementia may overestimate their functional abilities, and therefore it is important to consider the information provided by family members while diagnosing cases in the early stages of the disease [1]. There have been also reports that older adults with mild cognitive impairment (MCI) are less aware of changes in instrumental activities of daily living (IADL) due to reduced self-awareness [2,3]. Caregivers’ and especially family members’ reports on IADL have been found useful for the diagnosis of dementia [4].

Previous studies on family members of people with MCI and mild Alzheimer’s disease (AD) have identified the caregiver burden and its associated factors [5,6,7,8,9], family members’ perceptions of patients’ behavioral and psychological symptoms of dementia (BPSD) [10,11], and the association between activities of daily living (ADL) and BPSD from the family members’ perspective [12,13,14].

Studies show that complex IADL, such as management of money and medication, decline in the early stages of MCI and mild AD [15,16,17]. Additionally, our earlier survey revealed that functions related to the ability to use the telephone, shopping, cooking, housekeeping, managing finances, and managing medication decline after subjective memory complaints emerge [18]. Furthermore, several reports suggest that BPSD begin during the MCI stage, with the emergence of symptoms of depression, dysphoria, apathy, irritability, and sleep disturbances (in order of frequency) [19,20,21,22].

Previous studies have revealed that IADL independence declines and BPSD (apathy, depression) emerge from the MCI stage, and the association between the two has already been confirmed [23,24,25]. Additionally, studies have highlighted that MCI can lead to BPSD and a decline in IADL [14,25]. For example, it has also been shown that apathy is a predictor of complex ADL, especially in mild AD with a clinical dementia rating = 0.5 [26]. Apathy may appear even at the stage of subjective memory complaints, and it may be related to IADL, but the current situation is that there is no knowledge. 

On the other hand, family members who routinely support individuals with memory complaints are aware of the care recipient’s problems; therefore, they seek appropriate support from the early stages of dementia. There were studies that examined the association between IADL and BPSD in AD patients from the information provided by the caregiver [13,27]. However, few studies use family members’ reports of patients’ IADL abilities to examine the association between IADL and BPSD before the onset. The Process Analysis of Daily Activity for Dementia (PADA-D) developed by our group is an ADL evaluation method that can be used to analyze the impairment in each daily activity associated with cognitive decline at the process level [23]. This method allows us to derive an in-depth understanding of the status of ADL performance of individuals with memory complaints by collecting data from caregivers and family members.

The present study aimed to clarify the characteristics and relevance of IADL and BPSD in individuals with memory complaints, assessed from the perspective of family members by using the PADA-D. This study provides an understanding of family members’ perception of IADL and BPSD in community-dwelling older adults. Additionally, its findings may help provide support for affected individuals and their family members from the stage at which memory complaints and subsequent issues emerge.

## 2. Materials and Method

### 2.1. Study Design

This was a cross-sectional study using a self-administered questionnaire.

### 2.2. Ethical Considerations

This study was approved by the ethics board of Kagoshima University. The questionnaire explained to participants what the purpose of the research was and that the information collected would be kept confidential and used only for the purpose of this study. By answering the survey questions, the respondents indicated their agreement to participate. Our study protocol was approved by the Ethics Committee on Epidemiological Studies, Kagoshima University on 27 December 2018. Its identification code is 170377(370)-2.

### 2.3. Participant

Consumer’s Co-operative (CO-OP) is an autonomous association of consumers who volunteer to fulfill common needs and aspirations. CO-OP Kagoshima is a private enterprise with deep ties to the local community; it assists the residents with various activities, including conducting trips to stores, delivery, and other benefits that may support functioning. At CO-OP Kagoshima, many employees in stores and delivery businesses struggle to communicate with people with dementia or suspected dementia. In December 2018, a self-administered questionnaire was sent to 2000 randomly selected members of CO-OP Kagoshima who were 60 years old and over. We received 621 responses (recovery rate: 31%). The questionnaire included an enclosed reply envelope; thus, the responses were collected via mail. The survey period was from December 2018 to January 2019. Of the returned responses, there were 159 participants (individuals with memory complaints’ family) who answered about individuals with memory complaints. The individuals with memory complaints in this study were the participants’ family, and who were worried about memory complaints. In addition, the participants answered yes to the question “Is there any family member who worried about memory complaints and feel inconveniences in life?”.

### 2.4. Method

#### 2.4.1. Procedure

The participants answered questions related to the following regarding the individuals with memory complaints: 1. characteristics of individuals with memory complaints, 2. IADL-8 performance of PADA-D (ability to use the telephone, shopping, cooking, housekeeping, use modes of transportation, laundry, managing finances, managing medication), 3. the short version of the 28-item Dementia Behavior Disturbance scale (DBD13) and 4. observation list for early signs of dementia (OLD).

#### 2.4.2. Measurements

##### Characteristics of Individuals with Memory Complaints

Characteristics of individuals with memory complaints data were collected in the questionnaire. Items assessed for age, gender, residence, living situation (living alone or with family, number of cohabitants), level of care needed (the Japanese long-term care insurance: LTCI) [28,29], and work status (working or not working). Other items assessed feeling of health (very good, good, bad, or very bad) as seen by the participants (objective), degree of pain, and hobby involvement (or hobbies).

##### PADA-D

We used PADA-D to evaluate the IADL of the individuals with memory complaints. PADA-D can divide the performance of daily living activities by those with cognitive decline into processes and analyze ADL in detail. Five occupational therapists and two dementia specialists collaborated to determine the processes and actions that make up the process (actions) to be included from the Physical Self-Maintenance Scale and Lawton IADL scale. The PADA-D consists of a total of 14 activity performances (6 basic ADL performances and 8 IADL performances). Each activity performance is divided into five processes and three actions (Table 1). As an example, Table 2 demonstrates how “Ability to use the telephone” is divided into processes and actions. Appendix A show processes and actions of the other IADL performance. In PADA-D, activities’ performances are arranged in a time series from the beginning to the end of an action, and it is possible to clearly indicate which process is impaired. The examiner judges the actions by “doing (YES)” and “not doing (NO),” granting points for “doing (YES).” Three points are available for 1 process, 15 points for 1 performance, and a total of 210 points for the 14 performances. The final scale with the items that were selected had high internal consistency and criterion validity (Cronbach’s α = 0.96) [30]

##### DBD13

We used the DBD13 to evaluate BPSD of the individuals with memory complaints. The DBD13 was created in Japan as a short version of the DBD [31] for easy evaluation of BPSD in the clinical scene. Thirteen items with a high frequency of appearance and large change due to intervention were extracted, and these have an extremely significant correlation (r = 0.96, *p* < 0.0001) with the DBD. In addition, the DBD13 was shown to be reliable and valid [32]. We rated them on a five-point scale, from 0 (not at all) to 4 (always). The higher the total score of the DBD 13, the more severe the obstacle, and the total score is 52 points.

##### OLD

We used the OLD to understand the initial symptoms of dementia, such as memory complaints. The OLD is an evaluation of an observation formula consisting of 12 items that are likely to appear early in AD. We evaluated them on a two-point scale, 1 (yes) or 2 (no). If more than four items are “yes”, dementia is suspected. The OLD was shown to be reliable and valid [33].

#### 2.4.3. Statistical Analysis

Once data were gathered, totaled, and averaged, characteristics of individuals with memory complaints were considered. We calculated the appearance rate of items in the DBD13 (the percentage of respondents who answered “sometimes”, “frequently”, or “always”). Next, the multiple regression analysis was performed after checking data with a normal distribution using the Shapiro–Wilk test. If a normal distribution was not confirmed, the Durbin–Watson test was performed to confirm the normality of the residuals. The stepwise multiple regression analysis was used to examine the association between living activities and BPSD. The dependent variable was PADA-D total score, the independent variables were item scores of the DBD13, and the covariates were age, gender, and OLD total score. Finally, the stepwise multiple regression analysis was used to examine in detail which items of BPSD were related to daily activities. The dependent variables were each total score of PADA-D 8 IADL performance, the independent variables were item scores of the DBD13, and the covariates were age, gender, and OLD total score. If answers were missing, they were treated as missing values. All statistical analyses were performed using IBM SPSS Statistics ver25.0, and the significance level was set at 5% or 1%.

## 3. Results

### 3.1. Characteristics of Individuals with Memory Complaints

Table 3 shows the characteristics of persons concerned. The average age was 84.4±8.8 years old. They were mostly women (62.3%), 66% of whom lived with family. The average PADA-D total score was 55.8 ± 40.9 points, the average OLD total score was 6.4 ± 4.1 points, and the median DBD 13 total score was 14.0 points. The appearance rates of sub-items in the DBD13 were No1. “Ask the same question repeatedly” (79.7%), No2. “Loses, misplaces, or hides things” (65.1%), No6. “Sleeps excessively during the day” (60.3%), and No3. “Lack of interest in daily activities” (58.3%) in descending order. Other features were as follows: 55% looked healthy. The median level of care needed (LTCI) was Requiring Support 2 (a condition in which the ability to perform IADL was slightly reduced and some support was needed to maintain and improve functions). A total of 88% had no work, 66% had objective pain, and 31% had hobbies.

### 3.2. PADA-D Total Score and DBD13 Sub-Item Score

Table 4 shows the results of the stepwise multiple regression analysis of PADA-D total score and DBD13 sub-item score. PADA-D total score was associated with No3. “Lack of interest in daily activities” (β = −0.262, *p* = 0.007), No4. “Wake up at night for no obvious reason” (β = −0.277, *p* = 0.003), and No5. “Makes unwarranted accusations” (β = 0.193, *p* = 0.017) in the DBD13.

### 3.3. PADA-D 8 IADL Performance and DBD13 Sub-Item Score

Table 5 shows the results of the stepwise multiple regression analysis of each total score of PADA-D 8 IADL performance and DBD13 sub-item score. Many of the IADL were associated with multiple BPSD such as apathy, nocturnal wakefulness, and dresses inappropriately. 

#### 3.3.1. Ability to Use the Telephone

This was associated with No3. “Lack of interest in daily activities” (β = −0.36, *p* < 0.001), No4. “Wake up at night for no obvious reason” (β = −0.32, *p* < 0.001), No10. “Dresses inappropriately” (β = −0.02, *p* = 0.007), and No12. “Hoard things for no obvious reason” (β = 0.19, *p* = 0.005) in the DBD13.

#### 3.3.2. Shopping

This was associated with No3. “Lack of interest in daily activities” (β = −0.25, *p* = 0.006) and No4. “Wake up at night for no obvious reason” (β = −0.19, *p* = 0.035) in the DBD13.

#### 3.3.3. Cooking

This was associated with No4. “Wake up at night for no obvious reason” (β = −0.29, *p* = 0.001) in the DBD13.

#### 3.3.4. Housekeeping

This was associated with No4. “Wake up at night for no obvious reason” (β = −0.29, *p* = 0.001) in the DBD13.

#### 3.3.5. Use modes of Transportation

This was associated with No1. “Ask the same question repeatedly” (β = −0.19, *p* = 0.014) and No3. “Lack of interest in daily activities” (β = −0.31, *p* < 0.001) in the DBD13.

#### 3.3.6. Laundry

This was associated with No4. “Wake up at night for no obvious reason” (β = −0.38, *p* < 0.001) and No5. “Makes unwarranted accusations” (β = −0.24, *p* = 0.008) in the DBD13.

#### 3.3.7. Managing Finances

This was associated with No4. “Wake up at night for no obvious reason” (β = −0.34, *p* < 0.001), No5. “Makes unwarranted accusations” (β = 0.16, *p* = 0.032), and No10. “Dresses inappropriately” (β = −0.20, *p* = 0.006) in the DBD13.

#### 3.3.8. Managing Medication

This was associated with No1. “Ask the same question repeatedly” (β = −0.17, *p* = 0.037), No3. “Lack of interest in daily activities” (β = −0.32, *p* < 0.001), No10. “Dresses inappropriately” (β = −0.28, *p* = 0.001), and No11. “Refuses to be helped with personal care” (β = 020, *p* = 0.010) in the DBD13.

## 4. Discussions

In this study, we investigated the association between IADL ability and BPSD in individuals experiencing memory complaints, from their family members’ perspective, by collecting detailed information about the daily activities of community-dwelling older adults belonging to CO-OP Kagoshima. We found that the daily activities of the individuals with memory complaints were associated with BPSD such as apathy, nocturnal wakefulness, and unwarranted accusations. Additionally, each IADL was associated with age and BPSD such as apathy, nocturnal wakefulness, and dresses inappropriately.

The individuals with memory complaints included in this study were aged over 80 years, the majority of them were female, and about half of them lived with someone else. Additionally, their ability to perform IADL was slightly low according to their level of care needed (Requiring Support 2), indicating that they needed some kind of support. Further, several of them experienced objective pain, muscle weakness, and hearing loss. The average total OLD scores suggested that some of them may have had early dementia. Further, their median DBD13 total score indicated that some of them may have experienced BPSD symptoms such as memory loss, losing items, sleeps excessively during the day, and apathy. We found that family members noticed memory changes and the emergence of BPSD.

According to previous studies on IADL and BPSD reported by family members of AD patients, the Functional Activities Questionnaire (FAQ) positively correlated with all items, expect for one (depression), in the Neuropsychiatric Inventory (NPI) [13]. Additionally, sleep disorders (nocturnal wakefulness) in AD patients were highly associated with anxiety levels (OR2.1) and ADL disorder (OR1.6) [12]. Apathy for normal and MCI participants was moderately associated with IADL [25]. Vascular dementia (VaD) patients’ level of apathy has also been reported as a predictor of BADL/IADL performance [34]. BPSD, such as depression, apathy, agitation, irritability, disinhibition, and anxiety, are also seen in older adults with cognitive impairment and no dementia (CIND), and these symptoms are reported to be associated with functional limitations in basic ADL and IADL [23]. In individuals with MCI, this suggests that changes in cognitive function may affect daily activity, and that depression and apathy may have a strong effect on everyday function [14]. As different scales are used to assess daily activities and BPSD, it is difficult to compare the present results with those of previous studies. However, the present finding of an association of IADL with apathy, nocturnal wakefulness, and unwarranted accusations is in line with the findings of previous studies mentioned above. For example, apathy disrupts lifestyle habits and affects overall life, including health and hygiene, and sleep disorders can hinder the execution of daily activities at appropriate times. Additionally, unwarranted accusations could make interpersonal relationships difficult and could lead to caregivers’ burden. These findings suggest a bidirectional association between IADL and BPSD before the onset.

Next, we examined the association between eight IADL and BPSD individually. Each ADL was associated with multiple BPSD, and most were associated with apathy, nocturnal wakefulness, and dresses inappropriately. This finding confirmed that BPSD may be linked to impaired IADL ability. Additionally, we clarified that shopping, cooking, housekeeping, laundry, managing finances, and managing medication were related to aging. We confirmed that these factors cause a decline in IADL ability.

For instance, “Laundry” in the PADA-D comprises the following five processes: “Put the laundry in the washing machine,” “Start the washing machine,” “Operate the dryer or find another effective means to dry the laundry,” “Take in and fold the laundry,” and “Put the clothes in the chest/closet.” The present findings revealed that “Laundry” was significantly associated with nocturnal wakefulness and unwarranted accusations. Nocturnal wakefulness may lead to reduced physical activity during the day [35] and circadian rhythm disorders [36], which may delay and impair the ability to dry and retrieve laundry. Additionally, unwarranted accusations could be thought of as a manifestation of stress in not being able to do what was previously easy [5]. It is possible that the motivation to carry out a daily activity such as laundry may decrease.

Similarly, “Ability to use the telephone” comprises the following five processes: “Call others,” “Talk on the phone,” “Hang up the phone,” “Notice the phone ring,” and” Answer and talk on the phone.” The present findings revealed that “Ability to use the telephone” was significantly associated with apathy, nocturnal wakefulness, dresses inappropriately, and hoarding things. Apathy was strongly associated with IADL in MCI patients [37], and it may interfere with the act of answering or getting the phone. Additionally, a previous study reported that hoarding things implied the presence of memory complaints [5]. This suggests that memory decline may lead to difficulties with memorizing a message or responding to conversations with callers.

The present study has several limitations. First, we did not collect information on individuals with memory complaints’ medical history of MCI or dementia. Based on the OLD results and background information, several of them could be inferred to be at the SMC or MCI stage. However, we cannot be sure. Second, we did not collect the caregivers’ background information. There may have been biases regarding their care recipients, but this was unclear because we did not collect information on the association between the caregivers and care recipients. Third, it may be difficult to generalize the findings given that the participants were members of CO-OP Kagoshima, who are more aware of health issues as compared to the general population. Forth, the recovery rate in this study was 31%. Some studies were investigating with a recovery rate of 30−40% for community-dwelling older adults [38,39]. The recovery rate of this study is considered to be rather low, so we will consider a methodology to increase the recovery rate in future studies. Finally, the fact that the participants’ educational background was not investigated is one of the defects of this study. This is significant as a low level of education has been reported to be a risk factor for MCI and AD [40].

Despite these limitations, the results of this study indicate that family members are aware of individuals with memory complaints’ decline in IADL abilities and BPSD, which are likely to appear in MCI and the early stages of dementia. Accordingly, the implementation of interventions to modify the lifestyle of such individuals early, specifically when families recognize these changes, may help maintain and improve the long-term quality of life of care recipients and their family. Furthermore, this awareness needs to be taken into consideration during early medical consultations and appropriate support should be provided.

## 5. Conclusions

This study showed that the family members were well aware of individuals with memory complaints’ IADL and BPSD. Further, the results revealed that many of IADL were associated with multiple BPSD, particularly apathy and nocturnal wakefulness. We consider that investigating IADL and BPAD from a family who can observe the individuals with memory complaints in detail by a specific evaluation (like PADA-D) could help in accurately understanding the behavioral changes of them. In addition, bidirectional intervention of IADL and BPSD such as apathy and nocturnal wakefulness from the time the family notices behavioral changes may prevent progression to MCI and dementia and reduce the caregivers’ burden.

## Figures and Tables

**Table 1 ijerph-17-06831-t001:** List of processes of the PADA-D.

**IADL**	
**Performance**	**Process**
Ability to use the telephone	Call others
	Talk on the phone
	Hang up the phone
	Notice the phone ring
	Answer and talk on the phone
Shopping	Enter the store
	Go to the section
	Find a product
	Pay for the product
	Take home the product
Cooking	Plan a meal
	Prepare the food (wash, cut, and heat the ingredients)
	Season the ingredients (choose seasoning, etc.)
	Plate the food
	Set the table
Housekeeping	Clean up after a meal
	Managing daily necessities
	Management of bedding
	Clean the house
	Garbage dumping
Use modes of transportation	Take a taxi
	Take a bus or train
	Ride a bicycle
	Drive a mobility scooter
	Choose an appropriate mode of transportation
Laundry	Put the laundry in the washing machine
	Start the washing machine
	Operate the dryer or find another effective means to dry the laundry
	Take in and fold the laundry
	Put the clothes in the chest/closet
Managing finances	Handle cash
	Use cash on a daily life
	Understand household expenses
	Use the bank and the post office
	Use electronic money
Managing medication	Keep the regular time to take medication
	Take out the prescribed medicine
	Check the correct quantity of medicine
	Take medicine correctly
	Keep track of leftover medicine
**BADL**	
**Performance**	**Process**
Toileting	Get into the restroom
	Sit on the toilet seat
	Excreting
	Do post-processing
	Leave the restroom
Feeding	Choose a dish
	Shape the food easy to eat
	Bring food to the mouth
	Eating
	Finish the meal
Dressing	Choose clothes
	Take off clothes
	Put on clothes
	Put on and take off socks
	Wear and take off shoes
Grooming	Brush teeth
	Wash the face
	Shave/Put on makeup
	Dress hair
	Cut nails
Mobility	Get up
	Move around in one room
	Move between one room and others
	Go out of the house
	Go to neighborhood
Bathing	Take off clothes
	Pouring hot water over your body
	Soak in the bath
	Wash body and hair
	Dry off hair and body

PADA-D: Process Analysis of Daily Activity for Dementia. IADL: instrumental activities of daily living. BADL: basic activities of daily living.

**Table 2 ijerph-17-06831-t002:** Example of the PADA-D (ability to use the telephone).

Score	Process	Actions That Make up the Process	Check		Remarks
	1. Call others	(1) Pick up the phone	YES	NO	
		(2) Press or touch the call button	YES	NO	
		(3) Call the number (registered button, etc.)	YES	NO	
	2. Talk on the phone	(1) Hold the phone on ear	YES	NO	
		(2) Confirm the conversation partner	YES	NO	
		(3) Tell the matter	YES	NO	
	3. Hang up the phone	(1) End the conversation	YES	NO	
		(2) Hold the phone away from the ear	YES	NO	
		(3) Press the call end button	YES	NO	
	4. Notice the phone ring	(1) Find the phone	YES	NO	
		(2) Confirm who is talking on the phone	YES	NO	
		(3) Press the call button	YES	NO	
	5. Answer and talk on the phone	(1) Hold the phone on ear	YES	NO	
		(2) Ask the purpose on the phone	YES	NO	
		(3) Have a conversation with each other	YES	NO	

Enter the score for each process in the score column after checking “YES” or “NO” for the three actions. We changed some words in the PADA-D we developed.

**Table 3 ijerph-17-06831-t003:** Characteristics of individuals with memory complaints.

Parameter	Individuals with Memory Complaints (*n* = 159)
age, mean (SD), year	84.4 (8.8)
female	99 (62.3)
living with family	107 (66.0)
PADA-D IADL total score (SD), max 120	55.8 (40.9)
OLD total score (SD)	6.5 (4.1)
DBD13 total score median (25th–75th percentiles)	14.0 (7.0–21.5)
DBD13 items appearance rate (sometimes–always)	
No1. Ask the same question repeatedly	79.7
No2. Loses, misplaces, or hides things	65.1
No3. Lack of interest in daily activities	58.3
No4. Wake up at night for no obvious reason	27.3
No5. Makes unwarranted accusations	24.5
No6. Sleeps excessively during the day	60.3
No7. Paces up and down	13.1
No8. Repeats the same action over and over	18.4
No9. Is verbally abusive, curse	19.6
No10. Dresses inappropriately	25.9
No11. Refuses to be helped with personal care	24.0
No12. Hoards things for no obvious reason	23.2
No13. Empties drawers or closets	11.6

Data are presented as *n* (%). PADA-D: Process Analysis of Daily Activity for Dementia. OLD: observation list for early signs of dementia. DBD: Dementia Behavior Disturbance scale.

**Table 4 ijerph-17-06831-t004:** The stepwise multiple regression analysis between PADA-D total score and DBD13 items.

Parameter	B	β	*p*-Value	95% CI, Lower	95% CI, Upper
No.3 Lack of interest in daily activities	−7.675	−0.262	0.007	−13.247	−2.108
No.4 Wake up at night for no obvious reason	−8.763	−0.277	0.003	−14.562	−2.964
No.5 Makes unwarranted accusations	7.149	0.193	0.017	1.283	13.016

β: regression coefficients. CI: confidence intervals. PADA-D: Process Analysis of Daily Activity for Dementia. DBD: Dementia Behavior Disturbance scale. Adjusted R^2^: 0.49. Covariate: age (β = −0.24, *p* = 0.002), OLD total score (β = −0.26, *p* = 0.002).

**Table 5 ijerph-17-06831-t005:** Multiple regression analysis between PADA-D IADL performance and DBD13 items.

Parameter		PADA-D IADL Performance							
		Ability to use the Telephone	Shopping	Cooking	Housekeeping	Use Modes of Transportation	Laundry	Managing Finances	Managing Medication
DBD13 item	No1. Ask the same question repeatedly					−0.19 (−1.46, −0.17) *			−0.17 (−1.44, −0.05) *
	No2. Loses, misplaces, or hides things								
	No3. Lack of interest in daily activities	−0.36 (−1.95, −0.76) **	−0.25 (−2.13, −0.36) *			−0.31 (−1.86, −0.68) **			−0.32 (−1.98, −0.66) **
	No4. Wake up at night for no obvious reason	−0.32 (−1.91, −0.67) **	−0.19 (−1.99, −0.08) *	−0.29 (−2.39, −0.58) *	−0.29 (−2.01, −0.52) *		−0.38 (−2.84, −1.01) **	−0.34 (−2.01, −0.78) **	
	No5. Makes unwarranted accusations						0.24 (0.39, 2.47) *	0.16 (0.07, 1.51) *	
	No6. Sleeps excessively during the day								
	No7. Paces up and down								
	No8. Repeats the same action over and over								
	No9. Is verbally abusive, curse								
	No10. Dresses inappropriately	−0.20 (−1.76, −0.29) *						−0.20 (−1.83, −0.32) *	−0.28 (−2.49, −0.66) *
	No11. Refuses to be helped with personal care								0.20 (0.28, 2.05) *
	No12. Hoards things for no obvious reason	0.19 (0.28, 1.55) *							
	No13. Empties drawers or closets								
age			−0.38 (−0.37, −0.16) **	−0.21 (−0.27, −0.04) *	−0.17 (−0.20,−0.01) *		−0.17 (−022, −0.01) *	−0.22 (−0.19, −0.05) *	−0.22 (−0.22, −0.04) *
gender		−0.14 (−2.67, −0.17) *		−0.19 (−0.55,−0.03) *		−0.30 (−5.01, −1.58) **			
OLD					−0.24 (−0.56,−0.09) *		−0.23 (−0.59, −0.08) *	−0.31 (−0.57, −0.21) **	
Adjusted R^2^		0.51	0.40	0.25	0.27	0.44	0.26	0.51	0.41

Data were presented as regression coefficients β (95% confidence intervals. CI lower, upper). *: *p* < 0.05. **: *p* < 0.001. PADA-D: Process Analysis of Daily Activity for Dementia. DBD: Dementia Behavior Disturbance scale, OLD: observation list for early signs of dementia.

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
