# Peer review of "Association between Daily Activities and Behavioral and Psychological Symptoms of Dementia in Community-Dwelling Older Adults with Memory Complaints by Their Families"

_ijerph, 2020, doi:10.3390/ijerph17186831_

Round 1
Reviewer 1 Report
Overall impression: This is an important subject which has significant clinical relevance. With the rise in aging population globally, it is helpful to have more ways of tracking early signs of mild cognitive impairment and dementia. There were no ethical concerns raised. While the findings are interesting, the discussion and conclusions are not coherently woven around the findings. The following are some of my suggestions, which I hope will be helpful for the authors.
While DBD scale is readily available, DBD 13 could not be found. The citation # 23 Machida, A. Estimation of the reliability and validity of the short version of the 28-item Dementia Behavior Disturbance scale. Nippon Ronen Igakkai Zasshi 2012, 49, 463–467 was not fully accessible. Only the abstract was available in English. It would be helpful to understand a bit more about the rationale for why the 13 items were chosen for DBD13, as compared to the 28 items in the original version. Lines 246-248: “However, the present finding of association of IADL with memory loss, sleep disorders, and dressing inappropriately is in line with findings of previous studies mentioned above”. The DBD items “Ask the same question repeatedly”, “Wake up at night for no obvious reason” and “Dresses inappropriately” are referred to as “memory loss, sleep disorders and dressing inappropriately” in the discussion. This is a bit misleading, since these DBD items are really highlighting some of the behavioral disturbances of dementia, and do not necessarily translate into “memory loss” or the broader “sleep disorders”. “Nocturnal wakefulness” is used elsewhere, which is more specific.
Overall, the discussion needs to be flushed out more, making a case for why behavioral changes which appear early can be readily elucidated from detailed history obtained from caregivers, etc. The behavioral changes such as asking the same question repeatedly likely will be noticed by family members first. The following citation expands on just this one symptom: Hamdy RC, Lewis JV, Copeland R, et al. Repetitive Questioning Exasperates Caregivers. Gerontol Geriatr Med. 2018;4:2333721417738915. Published 2018 Jan 9. doi:10.1177/2333721417738915
The conclusion needs more information and further editing. For example, lines 295-6 “Further, information from family members indicated that nocturnal wakening, and dresses inappropriately” is clearly incomplete.
Author Response
Reviewer 1’s comments and suggestions
- While DBD scale is readily available, DBD 13 could not be found. It would be helpful to understand a bit more about the rationale for why the 13 items were chosen for DBD13, as compared to the 28 items in the original version.
Response; Thank you for raising important question. DBD13 was created in Japan as a shorted version of DBD for easy evaluation of BPSD in clinical scene. DBD is evaluated at the first visit and on average one year later, and 13 items with high frequency of appearance and large change due to intervention were extracted. These have an extremely correlation (r=0.96, p<0.0001) with the DBD28.
We chose DBD 13 because it reduces the burden of family for research and is highly reliable.
We have supplemented the “Materials and Method” section with additional explanations (lines147- 150).
- The items of DBD13 were expressed in misleading words. “Ask the same question repeatedly”, “Wake up at night for no obvious reason” and “Dresses inappropriately” are referred to as “memory loss, sleep disorders and dressing inappropriately” in the discussion.
Response; Thank you for this suggestion. And, thank you for sharing the citation on "asking the same question". In addition to your suggestions, other studies (Hawkev,k et al, 2005. Navarro,R et al, 2011) have shown how repetitive questions contribute to family stress and reduce those repetitive questions. However, as a result of re-analysis as pointed out by Reviewer 3, the item "Ask the same question repeatedly" was not so related to IADL. Additionally, we paraphrased “Wake up at night for no obvious reason “as “nocturnal wakefulness”.
- Overall, the discussion needs to be flushed out more, making a case for why behavioral changes which appear early can be readily elucidated from detailed history obtained from caregivers, et al.
Response; Thank you for this suggestion. As a result of our re-analysis, IALD was associated with BPSD such as apathy, nocturnal wakefulness and unwarranted accusations. There was a change in the results, but I added some citation and recreated the discussion.
There have been also reports that individuals with MCI were less aware of changes in IADL due to reduces self-awareness (Maria, S, 2015, Kayla, S, et al,2019) We consider that detailed objective information from families based on a specific evaluation method (like PADA-D) is credible and useful for understanding behavioral changes of individuals with memory complaints.
- The conclusion needs more information and further editing.
Response; Thank you for this suggestion. We reviewed the conclusions and rewrote them as follows (lines314 - 321).
“This study showed that the family members were well aware of individuals with memory complaints’ IADL and BPSD. Further, the results revealed that many of IADL were associated with multiple BPSD, particularly apathy and nocturnal wakefulness. We consider that investigating IADL and BPSD by family who can observe the individuals with memory complaints in detail by a specific evaluation (like PADA-D) could accurately understand of the behavioral changes of them. In addition, bidirectional intervention of IADL and BPSD such as apathy and nocturnal wakefulness from the time the family understand behavioral changes may prevent progression to MCI and dementia and reduce the caregivers’ burden.”
Reviewer 2 Report
The study was well presented and have several merits, however, the generalization of these findings are marred by the limitations some of which are mentioned in the discussion. Beyond that, I have only a minor concern i.e. the education background of the participants was not provided if the authors don’t have access to that information, it should be listed as a limitation.
Author Response
Reviewer 2’s comments and suggestions
The education background of the participants was not provided if the authors don’t have access to that information, it should be listed as a limitation.
Response; Thank you for raising important question. Mehlika, A, et al (2020) shows that low education may be a marker for the transition from MCI to AD, which is an important factor in early and mild stages such as MCI. This should have been investigated, but it was not in this study. It was considered to be one of the defects of this research and added to the “Limitation” section (lines 302- 305) with the following contents.
“Finally, the fact that the participants’ educational background was not investigated is one of the defects of this study. This is significant as low level of education has been reported to be a risk factor for MCI to AD [40]. “
Reviewer 3 Report
This is an interesting and well-written paper. However, I have some suggestions and comments which might improve the quality of this paper.
Introduction: Line 51- Change relationship to association - please change this throughout the text
Statistics: Why was a forced entry method used and not a stepwise method?
The presentation of the results is confusing. It is not clear which are the main predictors of IADL and BPSD. Figures with observed data and predictors would be helpful.
The conclusion is a repetition of the last paragraph.
Please discuss future studies.
Author Response
Reviewer 3’s comments and suggestions
- Introduction: Line 51- Change relationship to association - please change this throughout the text
Response; Thank you for this suggestion. I Changed the “relationship” to association” to fit the context throughout.
- Statistics: Why was a forced entry method used and not a stepwise method?
Response; Thank you for this suggestion. Because the independent variable was decided, we used the forced input method, but as you pointed out, we thought that stepwise was appropriate and re-analyzed it.  Since the statistical analysis method and results have changed, "Materials and Method" section (Lines165,168), "results" section (Lines190-229, Table4, Table5), "Discussion" section "(Lines 233-286) and" The Conclusions "section (Lines 313-320) has been modified, and the" Abstract "(Lines 32-36) has been modified accordingly.
- The presentation of the results is confusing. It is not clear which are the main predictors of IADL and BPSD. Figures with observed data and predictors would be helpful.
Response; Thank you for question and this suggestion. We consider that the main result of this study is the association between IADL and BPSD, such as apathy, nocturnal wakefulness and unwarranted accusations. The association between each IADL and the items of DBD13 was shown in Table 5, but it was shown in the figure (please see the attached for the figure) as suggested. We think that it is difficult to understand the detailed causal association of each element from the results of this study. We believe that future research is needed to examine the detailed causal association. In addition, we have added the following paragraph to the “result” section (lines200 - 201).
“Many of IADL were associated with multiple BPSD such as apathy, nocturnal wakefulness, dresses inappropriately. “
- The conclusion is a repetition of the last paragraph.
Response; Thank you for this suggestion. We reviewed the conclusions and rewrote them as follows (lines313 - 320).
“This study showed that the family members were well aware of individuals with memory complaints’ IADL and BPSD. Further, the results revealed that many of IADL were associated with multiple BPSD, particularly apathy and nocturnal wakefulness. We consider that investigating IADL and BPAD from a family who can observe the individuals with memory complaints in detail by a specific evaluation (like PADA-D) could accurately understand of the behavioral changes of them. In addition, bidirectional intervention of IADL and BPSD such as apathy and nocturnal wakefulness from the time the family understand behavioral changes may prevent progression to MCI and dementia and reduce the caregivers’ burden.”

Reviewer 4 Report
Thank you for the opportunity to review this manuscript. While I believe the topic is an important one. It is with some merits for this journal . However, it requires some major revisions.
And in Page 2/14, Lines 63-65, the authors mention “Few studies use family members’ reports of patients’ IADL abilities to examine the relationship between IADL and BPSD before the onset or in the early stages of dementia.” Is it the real situation, or is it because the authors did not give a comprehensive review? Authors should add some references to state the “few” (No previous studies address this issue? Or this issue has not been sufficiently solved? A clear statement of research problem should be provided), in order to clearly state the research problem or knowledge gap. In addition, authors also should provide more sufficient critical literature review to indicate the drawbacks of existed approaches for investigating the relationship daily activities and behavioral and psychological symptoms of dementia
The response rate (31%) may be a bit low.
Author Response
Reviewer 4’s comments and suggestions
- “Few studies use family members’ reports of patients’ IADL abilities to examine the relationship between IADL and BPSD before the onset or in the early stages of dementia.” IIs it the real situation, or is it because the authors did not give a comprehensive review? Authors should add some references to state the “few”
Response: Thank you for this question. It has been reported that cognitive function (memory, executive function, et al.) and BPSD are associated with IADL in MCI and mild AD(Rachel, B,et al,2018. Tekin, S, et al, 2011. Christine, Y,et al, 2015.). For example, it has also been shown that apathy is a predictor of complex ADL, especially in mild AD with clinical dementia rating=0.50(Carolina,D,et al, 2019). Apathy may appear even at the stage of subjective memory complaints, and it may be related to IADL, but the current situation is that there is no knowledge. On the other hand, in the above studies, the evaluation was mainly for the patients, and many caregivers were investigating BPSD for the patients and the caregiver burden. However, there have been also report that individuals with MCI are less aware of changes in IADL due to reduces self-awareness (Maria, S, et al,2015. Kayla, S, et al, 2019). We consider that this study is novel that it investigated the IADL of the individuals with memory complaints from the families’ information and examined the association with BPSD.
We have supplemented the “introduction” section with additional explanations (lines 46-71)
- The response rate (31%) may be a bit low.
Response; Thank you for this suggestion. Some previous studies had a questionnaire recovery rate of 30-40%(Reinie,C, et al, 2019. Shikimoto, R, et al, 2017). However, we consider that the recovery rate in this study was a little low, so we added this point to the “limitation” section (Lines 298-301) as follows.
“Forth, the recovery rate in this study was 31%. Some studies were investigating with recovery rate of 30-40% for the community-dwelling older adults [38,39]. The recovery rate of this study is considered to be rather low, so we will consider a methodology to increase the recovery rate in future studies.”
Round 2
Reviewer 4 Report
Authors have completely addressed all my concerns.
This manuscript is a resubmission of an earlier submission. The following is a list of the peer review reports and author responses from that submission.